# Administration of alendronate exacerbates ammonium chloride-induced acidosis in mice

**Mikayla Moody**[1], **Tannin A. Schmidt**[1], **Ruchir Trivedi**[2], **Alix Deymier**[1]*

1 Department of Biomedical Engineering, School of Dental Medicine, University of Connecticut Health Center, Farmington, CT, United States of America, 2 Department of Nephrology, School of Medicine, University of Connecticut Health Center, Farmington, CT, United States of America

* deymier@uchc.edu

## Abstract

Bone disease is highly prevalent in patients with chronic kidney disease (CKD), leading to an increased risk of bone fractures. This is due in part to metabolic acid-induced bone dissolution. Bisphosphonates (BPPs) are a potential treatment for inhibiting bone dissolution; however, there are limited studies observing the use of BPPs on acidotic patients. We aimed to determine efficacy of BPPs on maintaining bone health and pH regulation in acid-exposed mice. Using a diet-induced murine model of metabolic acidosis, we examined bone structure, composition, and mechanics as well as blood gases for three groups: control, acidosis, and acidosis + bisphosphonates (acidosis+BPP). Acidosis was induced for 14 days and alendronate was administered every 3 days for the acidosis+BPP group. The administration of BPP had little to no effect on bone structure, mechanics, and composition of the acidosis bones. However, administration of BPP did cause the mice to develop more severe acidosis than the acidosis only group. Overall, we discovered that BPPs may exacerbate acidosis symptoms by inhibiting the release of buffering ions from bone. Therefore, we propose that BPP administration should be carefully considered for those with CKD and that alkali supplementation could help minimize acidifying effects.

## Introduction

Bone disease is an extremely common comorbidity of chronic kidney disease (CKD) due to abnormal calcium-phosphate metabolism, changes in calcitriol and parathyroid hormonal levels, and metabolic acidosis [1]. This leads to increased fracture risk resulting in a 3–4 times higher mortality rate than that of the non-CKD population [2]. Thus, there is a great interest in treating CKD patients with osteoporosis medications, such as bisphosphonates (BPP), to maintain bone volume and quality, thus increasing overall health and quality of life. However, the prescription of BPP has been limited due to possible side effects, including increased kidney damage, risk of osteonecrosis of the jaw, and atypical femur fractures [3–6]. However, recent animal and human studies have shown that properly administered BPP does not appear to have significant negative effects on kidney function and instead improves bone health [6, 7].

org/10.6084/m9.figshare.22060427), and (https://doi.org/10.6084/m9.figshare.22060226).

**Funding:** AD had funding from her UConn Health Start-up fund and the NSF CAREER Grant (#2044870) (https://www.nsf.gov/awardsearch/showAward?AWD_ID=2044870&HistoricalAwards=false). MM received the UConn Harriott Fellowship (https://grad.uconn.edu/prospective-student/internal-awards/) and GEM Fellowship (https://www.gemfellowship.org/gem-fellowship-program/). The funders had no role in study design, data collection and analysis, decision to publish, or preparation of the manuscript.

**Competing interests:** The authors have declared that no competing interests exist.

Nonetheless, most of these studies fail to investigate the effects of BPP administration on patients that present with metabolic acidosis alongside CKD.

With reduced kidney function, CKD patients are less able to excrete acid and regulate serum bicarbonate $HCO_3^-$, leading to an accumulation of acid in the body. This reduction in body pH is associated with cardiac and immune dysfunction as well as a decrease in bone mineral density [8–11]. As the primary reservoir of buffering ions in the body, the skeleton undergoes dissolution during acidosis resulting in the release of buffering phosphate ($PO_4^{3-}$) and bicarbonate ($HCO_3^{2-}$) ions from the bone mineral. This process is mediated by a combination of physiochemical processes, where acid simply dissolves the mineral, and cell-mediated processes, where acid promotes osteoclastogenesis and cellular activity resulting in increased bone resorption [12–15]. Thanks to this release of buffering ions, the pH can be restored to more physiological levels, increasing patient health.

BPP acts primarily on cell-mediated processes by inhibiting osteoclast resorption, thus reducing bone loss and maintaining bone mass [16]. However, in cases of acidosis, this reduced bone dissolution may inhibit the release of buffering ions, exacerbating the acidosis and its associated complications. It is therefore necessary to understand how treatment with BPP affects acidosis and its consequences on bone health. In this study, we use a murine model of diet-induced acidosis to isolate the effects of BPP treatment on pH dysregulation and bone health.

## Materials and methods

### Induction of metabolic acidosis and administration of alendronate

4–5-month-old, wild-type, CD-1 male mice from Charles River Laboratory (Worcester, MA) were used in this study. All animal procedures were approved by the University of Connecticut Health Center (UConn Health) Institutional Animal Care and Use Committee (IACUC) via written consent and abided by the standards set by the National Institutes of Health [17]. The approved IACUC protocol number is AP-200306-1123. The mice were euthanized via carbon dioxide ($CO_2$) asphyxiation followed by cervical dislocation. No anesthesia methods were used in this study. The mice were separated into three groups (N = 8/experiment group): control, acidosis, and acidosis + bisphosphonate (acidosis+BPP) (Fig 1). Alendronate sodium trihydrate (7.5 µg/mL) (Sigma-Aldrich, #A4978) was administered through subcutaneous injections every three days at a dosage of 75 µg/kg (Fig 1C). Injections started 3 days before acidosis induction to ensure effectiveness. Acidosis was induced in the acidosis and acidosis+BPP groups through a graded dosing of ammonium chloride ($NH_4Cl$) in the drinking water, starting at a concentration of 0.2 M $NH_4Cl$ + 5% sucrose and ending at 0.6 M $NH_4Cl$ + 5% sucrose with increased increments of 0.1 M $NH_4Cl$ every three days (Fig 1B and 1C), as done previously [12, 13]. Like the acidosis and acidosis+BPP groups, the control group was also housed for fourteen days in the vivarium while the experiment was being conducted (Fig 1A). After 14 days of experimentation (acidosis induction), the mice were sacrificed and frozen at -20˚C.

### Blood and urine chemistries and assessment of acidosis and treatment

Approximately 300 µL of blood was collected from non-anesthetized mice via submandibular puncture procedures on days 0 and 14 of acidosis induction (N = 8 mice/experimental group). After blood collection, gauze was placed on the site of collection to stop the bleeding before the mouse was placed back in their cage. A Heska Epoc blood gas analyzer (Loveland, CO, USA), which is calibrated for each test card before a blood sample is inserted, was used to determine various metrics of the blood, including pH, bicarbonate ($HCO_3^-$), calcium ($Ca^{2+}$), potassium ($K^+$), and sodium ($Na^+$). Urine pH and mouse weights as well as food and liquid consumption

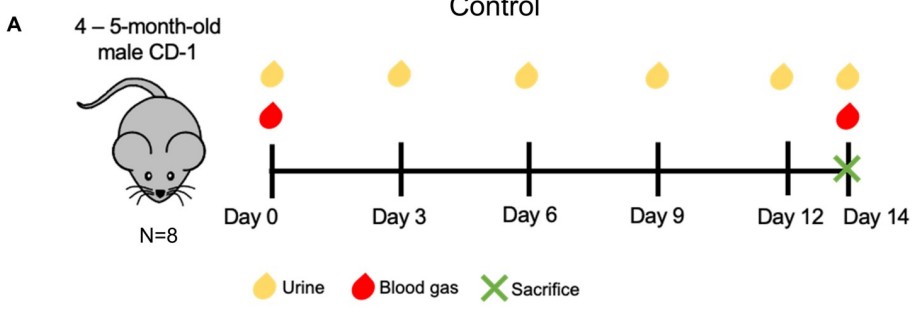

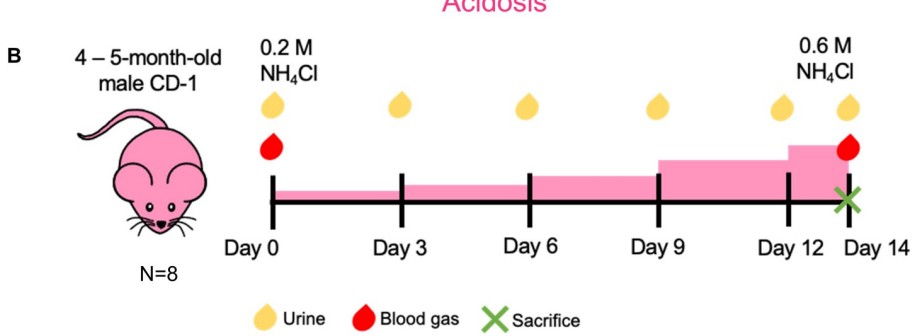

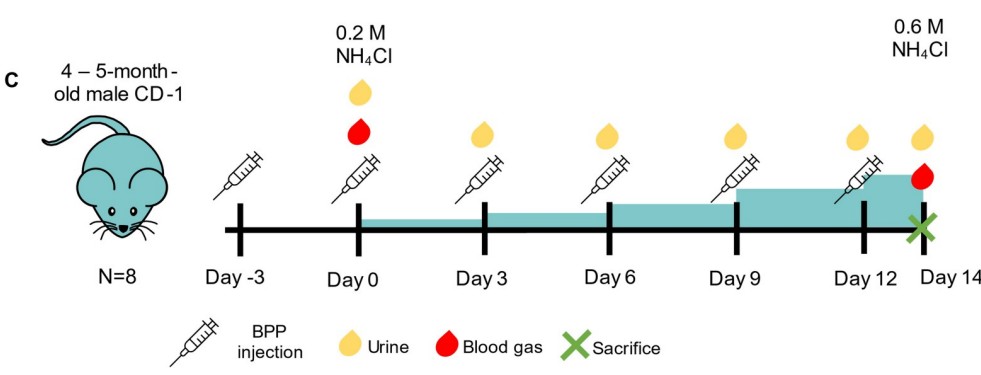

**Fig 1. Schematics for acidosis induction and alendronate administration.** The illustrations shown are for the (A) control, (B) acidosis, and (C) acidosis+BPP mice.

were measured every 3 days. Urine was collected via manual expression and its pH was measured using Hydrion pH strips (4.5–7.5) with a resolution of 0.5 pH units. Food consumption was measured by taking a baseline measurement and then weighing the food after three days. Liquid consumption was measured by taking a baseline measurement and then measuring the liquid after three days.

## Mechanical testing of bone

The mice were thawed, and left femurs were dissected on the day of testing. Three-point bend testing was done to measure the femur macroscale mechanics (N = 8 mice/experimental

group) using a Biomomentum (Laval, Canada) Mach-1 Mechanical Tester (v500csst with a 3-axis motion controller). During testing, the femurs were placed in a bath of 1X phosphate buffered saline (PBS) at 37˚C. The femurs were placed in a three-point bend rig (Biomomentum) with an 8 mm span and the posterior side facing up. They were then loaded at a rate of 0.1 mm/s until failure. Displacement was recorded using the stage displacement and the load was recorded with a 25 kg load cell. After mechanical testing, the femurs were wrapped in 1X PBS-soaked gauze and frozen at -20˚C. Load vs. displacement curves were used to determine structural mechanical properties, such as stiffness, maximum load, and yield load. The area moment of inertia and centroid distance as well as the span length were used to normalize the load and displacement values to calculate stress vs. strain curves. Custom MATLAB code was used to determine the material mechanical properties modulus, maximum stress, and toughness.

## Raman spectroscopy of bone

The composition of the femoral samples was measured using a Witec alpha 300 Raman spectrometer (Witec, Ulm, Germany) with a 785 nm laser at a grating of 300 g/mm. For exterior measurements, half of the femurs (N = 4 mice/experiment group) from mechanical testing were thawed at room temperature after being wrapped in 1X PBS-soaked gauze and preserved at -20˚C. Before taking measurements, the periosteum was removed using a scalpel blade and fine-grained sandpaper. Five point measurements were made on the exterior of the bones laterally to medially across the surface of the midshaft near the fracture site. The same femurs used for exterior measurements were used for interior measurements. Samples were not sectioned, but rather were turned 90˚ so that the internal surface exposed at the fracture site was facing the Raman laser. Six measurements were made across the distal fracture surface to obtain composition values for the bone interior: two spectra each on the lateral and medial sides and one spectrum each on the anterior and posterior sides. Each spectrum was taken with a 50X objective, center wavelength of 887.943 nm, laser power of 65 mW or less, and an acquisition time of 2 seconds x 30. The spectra were then cropped to 200–1800 $cm^{-1}$, background corrected via a rolling ball method with a ball size of 100 wavenumber, and cleared of cosmic rays using the Witec Program 5.3 software. The five exterior and the six interior spectra were each averaged together to create a single exterior and interior spectrum for each sample. These spectra were fit using a Lorentzian shape function in Witec Program 5.1 fitting software to obtain peak areas and full width at half maximums (FWHM) for the following peaks: the 960 $\Delta cm^{-1}$ v1 phosphate ($PO_4$) peak, the 1003 $\Delta cm^{-1}$ phenylalanine collagen peak, and the 1070 $\Delta cm^{-1}$ carbonate peak [18, 19]. The mineral:matrix ratio was determined from the ratio of the 960 $\Delta cm^{-1}$ and 1003 $\Delta cm^{-1}$ peak areas. The carbonate:phosphate ($CO_3^-$:$PO_4^{3-}$) ratio was determined from the ratio of the 1070 $\Delta cm^{-1}$ and 960 $\Delta cm^{-1}$ peak areas.

## Structural analysis of bone

After Raman spectroscopy analysis, all the left femurs used for mechanical testing were dehydrated in graded ethanol rinses up to 70% ethanol and imaged using a Scanco 50 microcomputed tomography (μCT) system (μCT 50, Scanco Medical, Bruttisellen, Switzerland) (N = 8/ experimental group). The scans were run at an energy of 55 kV and current of 145 μA with a Cu Kα X-ray source. Scans were done over an angular range of 180˚ with a step size of 0.36˚ (500 projections) and an acquisition time of 400 ms to obtain a 16 μm voxel size. The experimental data was used to determine the (1) fracture area parameters, (2) cortical parameters, and (3) trabecular parameters. 30 μCT sections on each side of the fracture were examined using BoneJ on ImageJ (U. S. National Institutes of Health, Bethesda, Maryland, USA) [20] to

measure centroid distance and area moment of inertia near the fracture site. The cortical thickness (Ct.Th) was determined by selecting a region of interest starting at the beginning of the fracture on the distal side and ending at 100 µCT sections going towards the condyles. Trabecular parameters were measured both in the distal epiphysis and in the distal diaphysis. For the distal epiphysis, µCT sections were selected between the start of the trabecular region on the distal end to the beginning of the growth plate. For the distal diaphysis, a region of interest containing 100 sections proximal from the end of the growth plate was selected. Using the Bruker CTan software, both the cortical and trabecular segmentation procedures included global thresholding, despeckling black spots less than 10 pixels from the images, ROI shrink-wrapping, and morphological operations. Thresholding values were selected by an experienced user after a preliminary examination of the data set and set to equal values across all samples. The trabecular parameters obtained were bone volume/total volume (BV/TV), trabecular thickness (Tb.Th), trabecular number (Tb.N), and trabecular spacing (Tb.Sp).

## Statistical analysis

Statistical analysis and graphical visualization were done in GraphPad Prism version 9.4.0. Day 0 is the day of acidosis induction right before giving the acidosis groups ammonium chloride. Day 14 is two weeks of graded dosing of ammonium chloride in acidosis drinking water, or for the control mice, two weeks of being maintained in the vivarium. Due to the normal distribution of data, parametric statistical testing using two-way ANOVAs or mixed-effects analysis with post-hoc Bonferroni's or Tukey's multiple comparisons tests were used for the blood metrics, food consumption, urine pH, weight, and fluid consumption. For the compositional, mechanical, and structural data, non-parametric statistical testing was used including Kruskal-Wallis tests with post-hoc Dunn's multiple comparisons tests.

## Results

### Blood gas, urine, and weight analysis

The blood gas parameters were measured at Days 0 and 14 for all groups. The acidosis+BPP group had a reduction in blood pH and $HCO_3^-$ between Days 0 and 14 (Fig 2A and 2B). However, the control and acidosis groups did not have any changes in these parameters. The acidosis and acidosis+BPP groups had an increase in blood $Ca^{2+}$, chlorine ($Cl^-$), and $Na^+$ at Day 14 compared to Day 0, while the control group stayed consistent over this time period (Fig 2C–2E). Moreover, there was an increase in glucose with the acidosis+BPP group, but no changes in blood urea nitrogen (BUN), as observed with the control and acidosis groups (Fig 3A and 3B). The acidosis group also had a decrease in lactate from Day 0 to Day 14 (Fig 3C). The base excess in blood (BE(b)) decreased in both the acidosis and acidosis+BPP groups (Fig 3D). There were no changes in partial pressure of carbon dioxide ($pCO_2$), partial pressure of oxygen ($pO_2$), and anion gap, $K^+$ (AgapK) within any of the groups over the span of 14 days (Fig 3E and 3G).

At Day 0, the urine pH was significantly higher in the groups with acidosis compared to the control; however, this changed once acidosis was induced. From Day 3 to Day 14, the urine pH in the acidosis and acidosis+BPP groups decreased compared to the control group. Moreover, on Day 6, the acidosis+BPP had a higher urine pH compared to the acidosis group. On the other hand, the acidosis group had a higher urine pH compared to the acidosis+BPP group on Day 12 (Fig 4A).

The animal weight between the three groups was similar throughout the 14 days. The acidosis+BPP groups had a trending increased weight compared to the acidosis group on Day 9

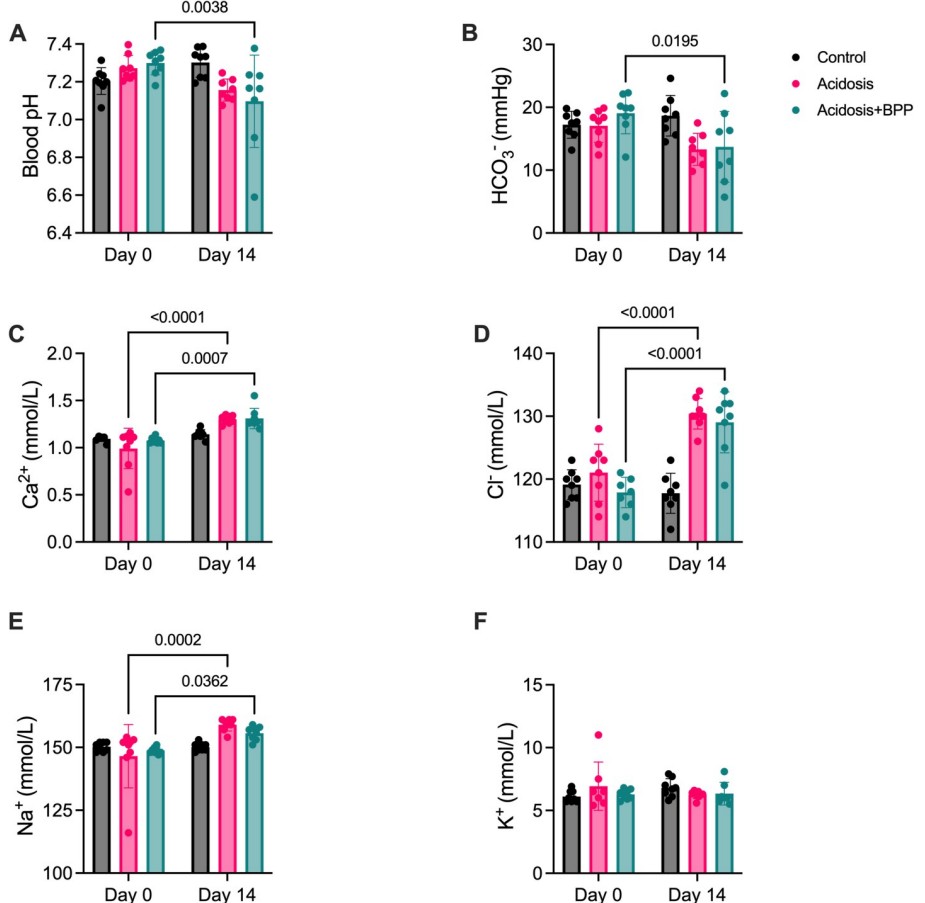

**Fig 2. Blood gas measurements of before acidosis induction (Day 0) and after 14 days of acidosis induction (Day 14) for the control, acidosis, and acidosis+BPP groups.** The measurements collected were (A) blood pH, (B) $HCO_3^-$, (C) $Ca^{2+}$, (D) $Cl^-$, (E) $Na^+$, and (F) $K^+$. For blood pH, $HCO_3^-$, $Ca^{2+}$, and $Na^+$, two-way repeated measures ANOVAs and post-hoc Bonferroni's tests were used. For $Cl^-$ and $K^+$, mixed-effects analysis and post-hoc Bonferroni's tests were used.

(p = 0.0832). On Days 12 and 14, the acidosis group had a trending smaller weight than the control (respectively, p = 0.0947 and p = 0.0771) (Fig 4B).

The groups undergoing acidosis were shown to drink and eat less. Days 3 through 14 showed a decrease in fluid consumption for the acidosis and acidosis+BPP groups compared to the control group (Fig 4C). At Day 3, the acidosis mice were already consuming less food than the control mice. However, this only continued for Day 6. On Days 6 and 9, the acidosis +BPP mice were eating less than the control mice (Fig 4D).

## Assessment of bone tissue composition in response to bisphosphonate and acidosis

There were no differences between the three groups when looking at the exterior and interior composition in terms of the mineral:matrix ratio, the $CO_3^-:PO_4^{3-}$ ratio, and the inverse of FWHM of the 960 $cm^{-1}$ peak (Fig 5).

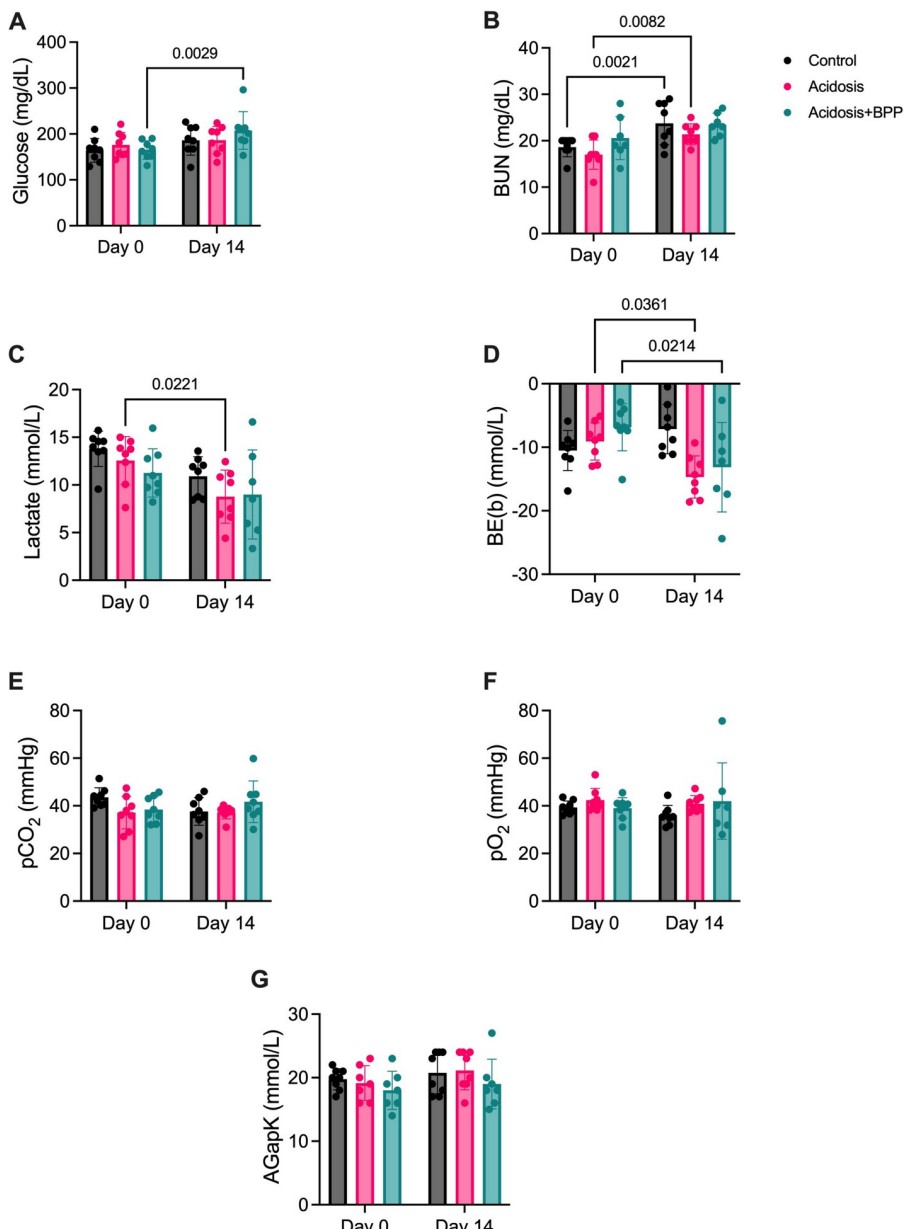

**Fig 3. Additional blood gas measurements of before acidosis induction (Day 0) and after 14 days of acidosis induction (Day 14) for the control, acidosis, and acidosis+BPP groups.** The measurements collected were (A) glucose, (B) BUN, (C) lactate, (D) BE(b), (E) $pCO_2$, (F) $pO_2$, and (G) AGapK. For glucose and $pCO_2$, two-way repeated measures ANOVAs and post-hoc Bonferroni's tests were used. For BUN, lactate, BE(b), $pO_2$, and AGapK, mixed-effects analysis and post-hoc Bonferroni's tests were used.

## Bisphosphonate administration and acidosis effects on whole bone femur mechanics

To determine how structural factors affect mechanical properties of the femurs in each group, the outer diameter and length were measured. The three groups had no differences for outer diameter (Fig 6A). However, after 14 days of acidosis, the acidosis+BPP group femurs had a trending increase in length compared to the control bones (p = 0.0811) (Fig 6B). The centroid

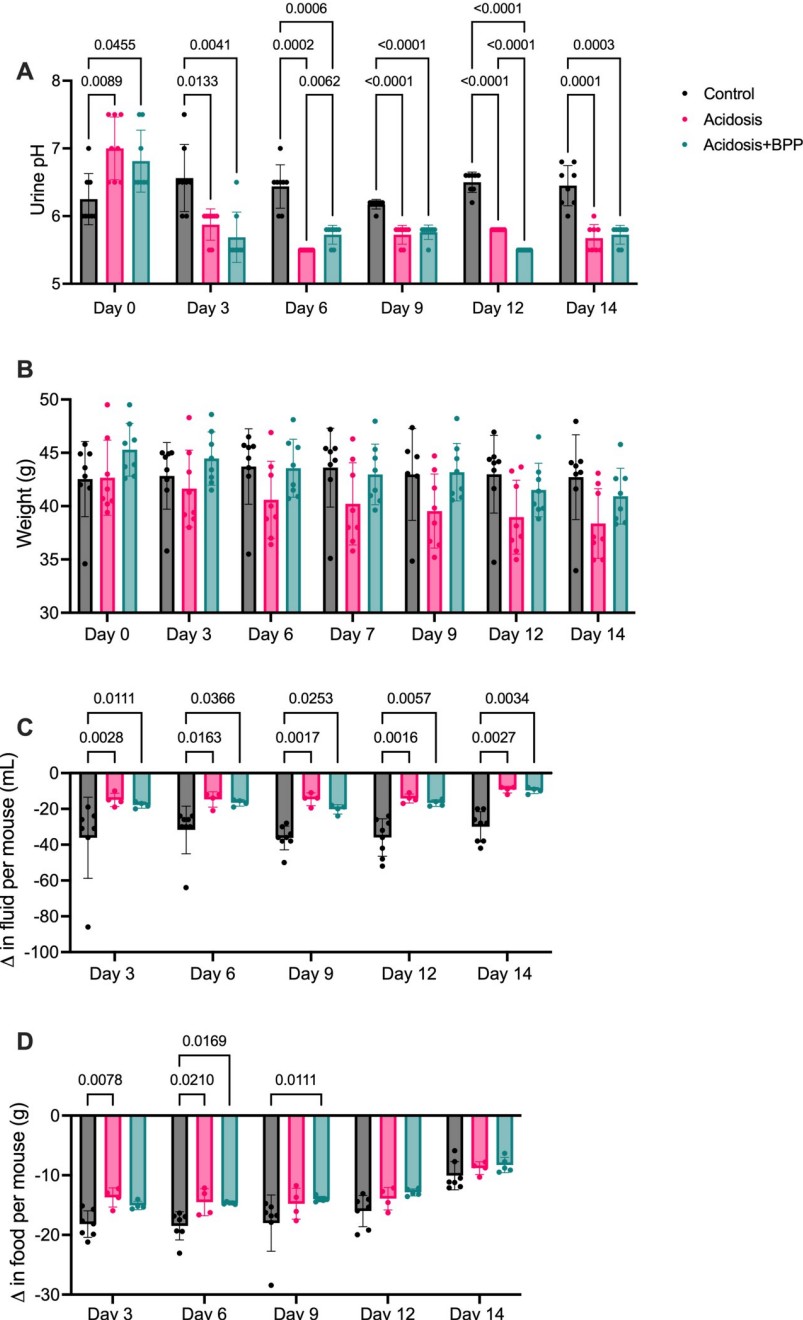

**Fig 4. A decrease in urine pH and food and water consumption over a span of 14 days of acidosis induction.** The measurements collected were (a) urine pH, (b) weight, (C) fluid consumption, and (D) food consumption. For urine pH, fluid consumption, and food consumption, two-way ANOVAs and post-hoc Tukey's tests were used. For weight, mixed-effects analysis and post-hoc Tukey's tests were used.

distance and area moment of inertia were also calculated from microCT data for mechanical influence (Fig 6C and 6D). For the area moment of inertia, the acidosis+BPP group was higher compared to the control group. For all the material and structural mechanical properties, there were no significant differences between the three groups (Fig 7).

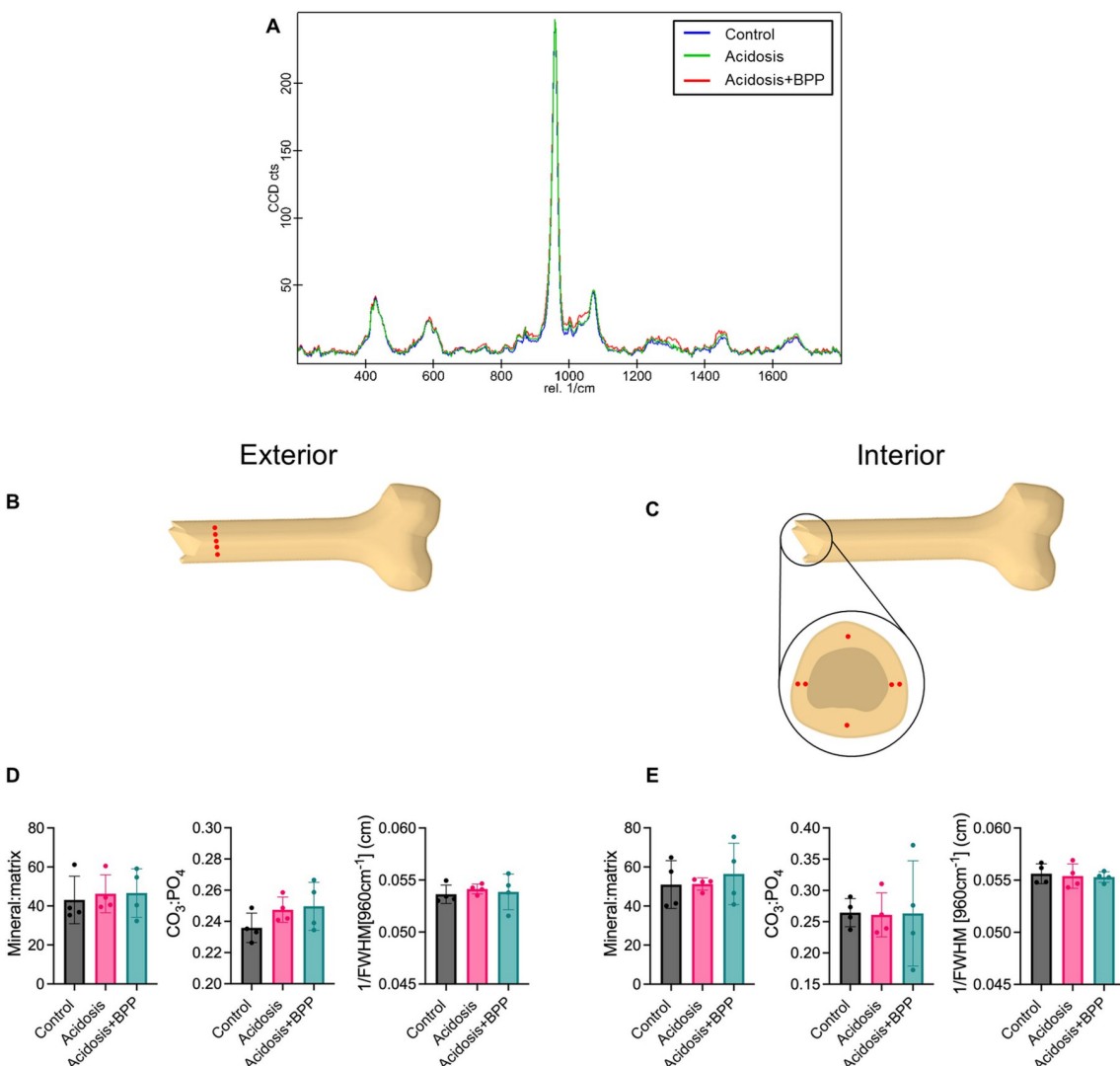

**Fig 5. No changes in composition via Raman spectroscopy after 14 days of acidosis induction.** (A) Raman spectra was used to measure composition on the (B) exterior and (C) interior of femur sample in order to determine the (D) exterior mineral:matrix ratio, the $CO_3$:$PO_4$ ratio, and the inverse of the FWHM of the 960 cm$^{-1}$ peak and the (E) interior mineral:matrix ratio, $CO_3$:$PO_4$ ratio, and the inverse of the FWHM of the 960 cm$^{-1}$ peak. For all data, Kruskal-Wallis tests with post-hoc Dunn's tests were used.

## Trabecular and cortical structural analysis in acidosis and bisphosphonate exposed bone

For the epiphysis, the trabecular BV/TV was significantly different, where the acidosis+BPP group had more bone volume than the control (Fig 8C). The diaphysis trabecular metrics had more significant differences between the groups. The acidosis+BPP group had a higher BV/TV and Tb.N and a lower Tb.Sp than the control (Fig 8F). Additionally, the acidosis group had a trending higher BV/TV ($p = 0.0549$) and significantly higher Tb.N and lower Tb.Sp than the control (Fig 8F). There were no changes in the Ct.Th among the three groups (Fig 8I).

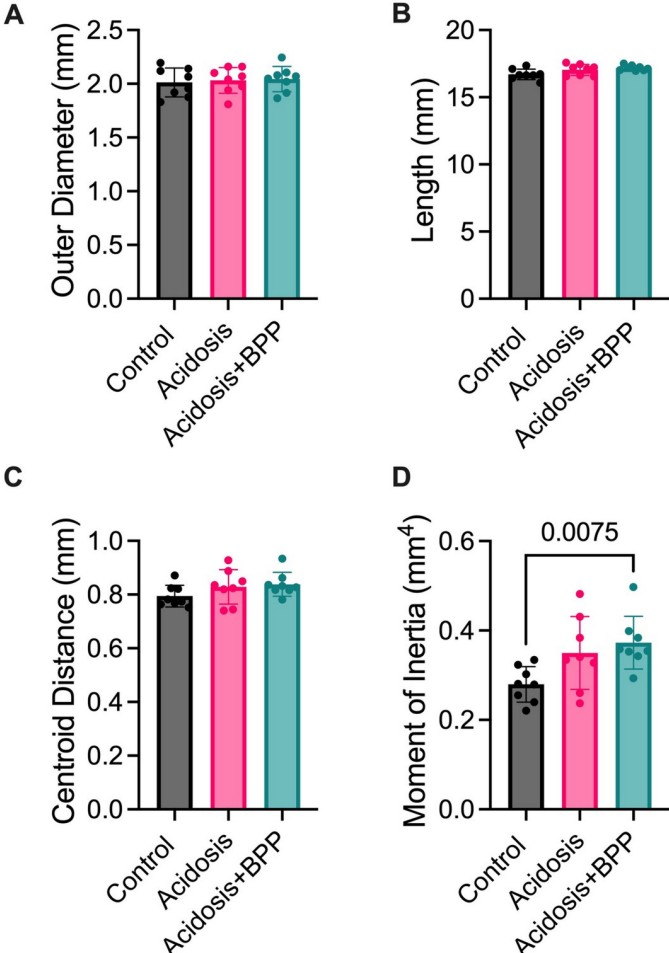

**Fig 6. Structural components that affect mechanical properties.** Gross measurements were taken for (A) outer diameter and (B) length of the femurs while microcomputed tomography was used to determine (C) centroid distance and (D) moment of inertia. For all data, Kruskal-Wallis tests with post-hoc Dunn's tests were used.

## Discussion

The relationship between BPP, acidosis, and bone health remains unexplored. However, there is a need to understand these interactions, especially with respect to the CKD population that presents with both bone disease, which could benefit from BPP treatment, and metabolic acidosis. Here we used an established murine model of diet-induced metabolic acidosis to isolate the effects of BPP on the pH dysregulation and bone health of the mice.

Much like CKD patients that exhibit acid accumulation but no reduction in blood pH or $HCO_3^-$, the acidosis mice in this study also presented with eubicarbonatemic or preclinical metabolic acidosis at Day 14 as expected for this model [12, 21–24]. Despite exhibiting pH and $HCO_3^-$ levels on par with Day 0 controls, the urine pH and Base Excess (BE(b)) were reduced with administration of the $NH_4Cl$, further supporting this hypothesis. However, the acidosis +BPP group showed a significant decrease in both pH and $HCO_3^-$ with 14 days of $NH_4Cl$ administration, suggesting that the alendronate treatment exacerbated the severity of acidosis in the mice. Like the acidosis group, the acidosis+BPP group also exhibited the reduced BE(b) and urine pH with $NH_4Cl$ administration.

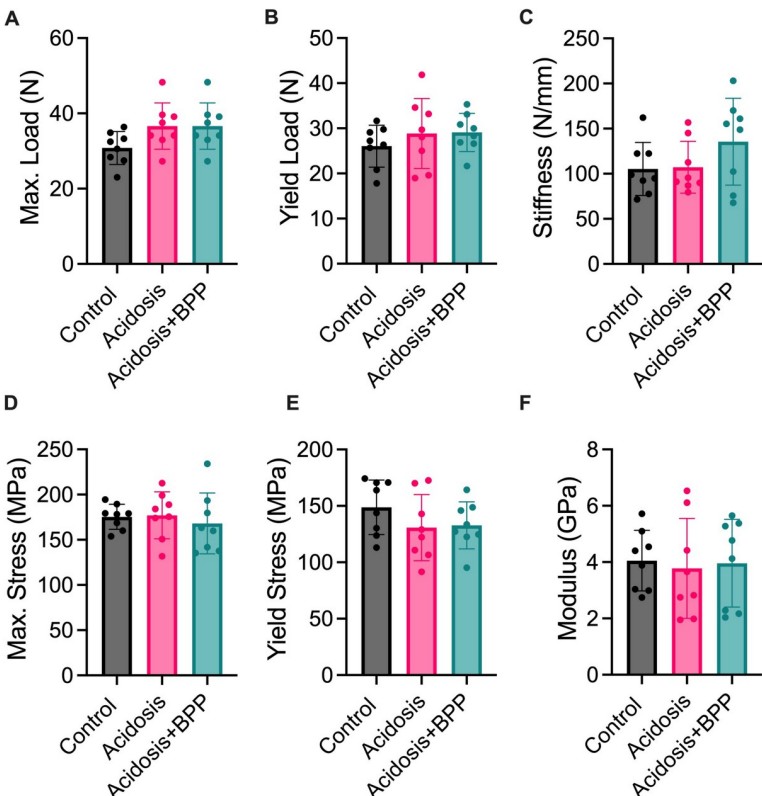

**Fig 7. No changes in mechanics via 3-point bend testing after 14 days of acidosis induction.** Structural mechanical properties measured were (A) maximum load, (B) yield load, and (C) stiffness. Material mechanical properties measured were (D) maximum stress, (E) yield stress, and (F) modulus. For all data, Kruskal-Wallis tests with post-hoc Dunn's tests were used.

There were no significant differences between the acidosis and acidosis+BPP groups in terms of food and water consumption, so the diet patterns of the mice should not have influenced the extent of acidosis experienced. The similar increases in blood $Cl^-$ also indicated that similar levels of $NH_4Cl$ were consumed by the two groups. This further suggested that alendronate exacerbated the metabolic acidosis, resulting in lower body pH and $HCO_3^-$ levels. This could be due to reduced osteoclast activity in the acidosis+BPP group, preventing the release of bicarbonate and other buffering agents from bone, leading to less buffering of the blood.

Interestingly, both the acidosis and the acidosis+BPP groups had an increase in circulating $Ca^{2+}$ and $Na^+$. Such an increase is generally used as an indicator of bone dissolution or increased remodeling [25, 26]. This would suggest that despite the administration of alendronate in the acidosis+BPP group, there is still a net release of ions from the bone material. To determine whether this release would have significant effects on the bone function, these bones were tested for compositional, structural, and mechanical changes.

The acidosis+BPP group exhibited an increase in BV/TV at both the epiphysis and the diaphysis at 14 days compared to controls. This was accompanied by an increase in trabecular number and a decrease in trabecular spacing in the diaphysis, suggesting that the alendronate induced a small increase in bone volume compared to the healthy control. Similar changes were seen in the acidosis group for the diaphysis; although, the extent of the change was larger for the acidosis+BPP group. The reason for the increase in bone volume with acidosis is unclear, but the greater accumulation of bone tissue in the epiphysis and diaphysis of the

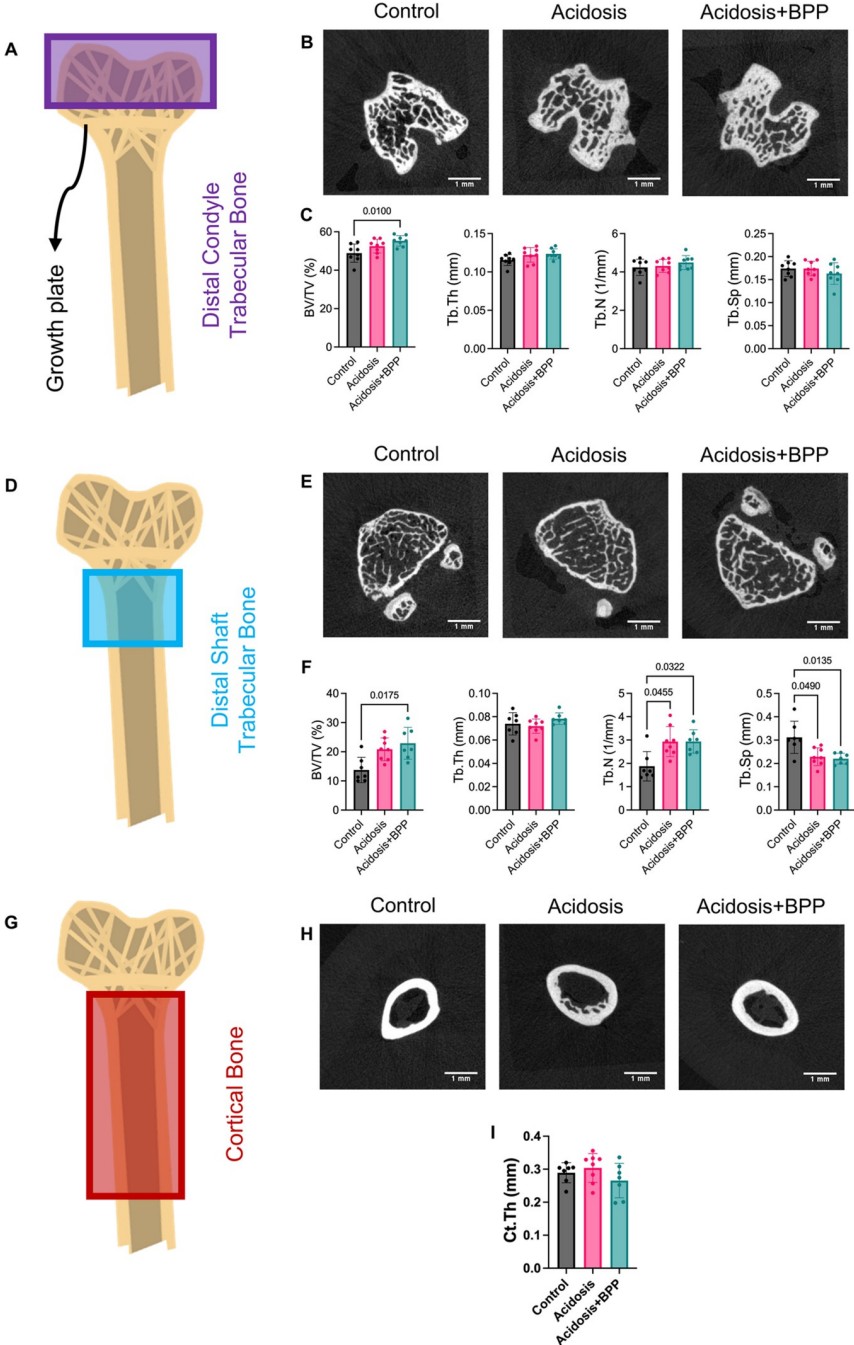

**Fig 8. BPP administration impacted distal diaphysis trabecular parameters after 14 days of acidosis induction.** In order to determine trabecular parameters in the (A) distal condyle, (B) microCT sections were used to examine (C) BV/TV, Tb.Th, Tb.N, and Tb.Sp. In order to determine trabecular parameters in the (D) distal shaft, (E) microCT sections were used to examine (F) BV/TV, Tb.Th, Tb.N, and Tb.Sp. In order to determine cortical parameters in the (G) shaft, (H) microCT sections were used to examine (I) Ct.Th. For all data, Kruskal-Wallis tests with post-hoc Dunn's tests were used.

acidosis+BPP group suggests that alendronate treatment led to increased bone deposition despite the measured increase in acidosis. This is in agreement with other studies showing an increase in trabecular volume with BPP treatment in rats with CKD, although their acidosis

status was not reported [7, 27]. Although there were structural changes to the bone with acidosis and acidosis+BPP, there were no measurable changes in the bone composition or bone mechanics at 14 days. In a previous study of our acidosis model, we saw changes in toughness between Days 1 and 14 [13]. Since this current study only looks at one time point, we might have only observed a recovery in the bone mechanics, which might be the reason for a lack in change after 14 days of acidosis [13].

Additionally, the blood gas results showed an increase in glucose in the acidosis+BPP groups over the span of 14 days. Although alendronate and other bisphosphonates have an association with decreasing insulin resistance, which in turn would decrease blood glucose [28, 29], acidosis has the opposite effect [30, 31]. Even though the acidosis+BPP group received alendronate, it seems that the effect of acidosis was greater than that of the acidosis group, potentially increasing glucose levels due to an increase in insulin resistance. We also saw a decrease in lactate in the acidosis group, which has been seen in previous studies [12]. However, the lactate might not have decreased in the acidosis+BPP group because there was a balance between the acidosis-mediated lactate lowering and the elevation of lactate levels due to a potential decrease in insulin resistance from the alendronate [32]. Together, these results suggest that when administering BPP to individuals with CKD and other co-morbidities, it is also essential to consider other systemic effects, such as insulin resistance, especially since CKD and diabetes are so often comorbidities [33].

The administration of BPPs could be a useful tool for the treatment of bone disease in individuals with CKD; however, the relationship between BPP, acidosis, and bone health remained unclear. Therefore, in this study, we used a model of diet-induced metabolic acidosis to isolate the effects of BPP on acid accumulation and bone health. We found that the administration of alendronate to mice undergoing acid dosing led to a decrease in blood pH and $HCO_3^-$ levels compared to acidotic mice, suggesting that BPPs could lead to increased acidosis. We hypothesize that this is a result of reduced osteoclast activity resulting in decreased bone resorption and thus decreased release of buffering ions. The administration of alendronate also had more impactful effects on bone volume, suggesting that alendronate does promote bone growth; however, this change was minimal compared to the acidosis group and had no effect on the composition or mechanics of the bones. Our data suggests the prescription of BPPs in individuals with acidosis should be a case-by-case basis with a balance of concerns about worsening acidosis and bone disease risks. However, acidosis in these scenarios could potentially be combated with the supplementation of alkali treatment [34]. Therefore, the dual administration of alkali treatment and BPP administration needs to be further explored.

## Acknowledgments

We would like to acknowledge the UCHC Micro CT Imaging Facility and specifically Renata Rydzik for assistance with the µCT data acquisition.

## Author Contributions

**Conceptualization:** Mikayla Moody, Alix Deymier.

**Formal analysis:** Mikayla Moody.

**Funding acquisition:** Alix Deymier.

**Investigation:** Mikayla Moody.

**Methodology:** Mikayla Moody.

**Project administration:** Alix Deymier.

**Resources:** Tannin A. Schmidt, Alix Deymier.

**Supervision:** Tannin A. Schmidt, Ruchir Trivedi, Alix Deymier.

**Visualization:** Mikayla Moody.

**Writing – original draft:** Mikayla Moody.

**Writing – review & editing:** Mikayla Moody, Tannin A. Schmidt, Ruchir Trivedi, Alix Deymier.

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
