## [Decision Letter · Decision Letter 0]

17 Jul 2023

PONE-D-23-12812Administration of alendronate exacerbates ammonium chloride-induced acidosis in micePLOS ONE

Dear Dr. Deymier,

Thank you for submitting your manuscript to PLOS ONE. After careful consideration, we feel that it has merit but does not fully meet PLOS ONE’s publication criteria as it currently stands. Therefore, we invite you to submit a revised version of the manuscript that addresses the points raised during the review process.

We look forward to receiving your revised manuscript.

Kind regards,

Furqan A. Shah

Academic Editor

PLOS ONE

Journal Requirements:

2. To comply with PLOS ONE submissions requirements, in your Methods section, please provide additional information regarding the experiments involving animals and ensure you have included details on (1) methods of anesthesia and/or analgesia, and (2) efforts to alleviate suffering.

4. Please expand the acronym “UConn, NSF” (as indicated in your financial disclosure) so that it states the name of your funders in full.

"AD had funding from her UConn Health Start-up fund and the NSF CAREER Grant (#2044870) (https://www.nsf.gov/awardsearch/showAward?AWD_ID=2044870&HistoricalAwards=false). MM received the UConn Harriott Fellowship (https://grad.uconn.edu/prospective-student/internal-awards/) and GEM Fellowship (https://www.gemfellowship.org/gem-fellowship-program/). The funders had no role in study design, data collection and analysis, decision to publish, or preparation of the manuscript."

We note that one or more of the authors is affiliated with the funding organization, indicating the funder may have had some role in the design, data collection, analysis or preparation of your manuscript for publication; in other words, the funder played an indirect role through the participation of the co-authors. If the funding organization did not play a role in the study design, data collection and analysis, decision to publish, or preparation of the manuscript and only provided financial support in the form of authors' salaries and/or research materials, please do the following:

(1) Review your statements relating to the author contributions, and ensure you have specifically and accurately indicated the role(s) that these authors had in your study. These amendments should be made in the online form.

(2) Confirm in your cover letter that you agree with the following statement, and we will change the online submission form on your behalf: 

Reviewers' comments:

Reviewer's Responses to Questions

**Comments to the Author**

1. Is the manuscript technically sound, and do the data support the conclusions?

Reviewer #1: Yes

Reviewer #2: Yes

2. Has the statistical analysis been performed appropriately and rigorously? 

Reviewer #1: Yes

Reviewer #2: No

3. Have the authors made all data underlying the findings in their manuscript fully available?

Reviewer #1: Yes

Reviewer #2: Yes

4. Is the manuscript presented in an intelligible fashion and written in standard English?

Reviewer #1: Yes

Reviewer #2: No

5. Review Comments to the Author

Reviewer #1: The article by Moody et al. reports a study on the role of bisphosphonates (BPP) (alendronate) in the recovery of acid-induced bone dissolution. With this aim the authors used a diet-induced murine model of metabolic acidosis (control vs acidosis vs acidosis+BPP; 8 mice / experimental group), to reproduce acid-induced bone loss in patients with chronic kidney disease, and they further characterized structure, mechanics, and composition of bones.

The data reported suggest that BPP, instead of reducing bone loss, induce a more severe acidosis, probably due to reduced osteoclast activity, preventing the release of bicarbonate and other buffering agents from bone.

The study is well conceived, and the reported data support the conclusions, even if they need to be further validated by specific studies focused on the mechanisms regulating the BPP-induced acidosis and on the impact of BPP on osteoclasts activity. The introduction clearly defines the biomedical question; the methods are clear, and figures are generally clear.

I have some comments on Raman experiments and associated results.

- in the manuscript it is not specified how bone samples have been prepared before Raman measurements. Were bone samples cut into sections? In case, which is the thickness, and which is the method used for cutting? Did author perform fixation other than the freezing step?

- I would add the type of grating used for the Raman measurements

- considering that the original measurements included spectral information from 200 and 1800 1/cm, I would not exclude to show the region between 200 and 960 1/cm, including two important phosphate bands, around 430 and 600 1/cm

- The use of mineral to matrix and carbonate to phosphate ratio are meaningful. At the same time, I would also consider crystallinity (inverse of the full-width at half maximum (FWHM) of the ν1PO4 (phosphate) peak at around 960 cm-1) as a parameter of bone quality. This could eventually provide interesting insights. From visual examination, a small difference can be appreciated at that level. If needed, here two good references: 1) https://doi.org/10.1007%2Fs11999-010-1692-y 2) https://doi.org/10.1039/D1AN01560E

Reviewer #2: The authors point-out and demonstrate the importance of careful consideration of the acidosis status of CKD patients when prescribing bisphosphonate therapy. While the negative of bisphosphonates on acidosis has been demonstrated, the manuscript would benefit from a major rewriting and additional investigations. Major revision is necessary.

Below is the list of concerns and suggestions for the authors in no particular order of priority.

1. Use of parametric tests in the statistical analysis with a small sample size (N=8 / group) should be avoided. Data should be re-analysed and non-parametric alternative analyses performed. Reporting of P values less than and equal to 0.1 in the graphs is misleading, especially in cases where both the significant and non-significant differences are reported. I suggest those to be removed.

2. Spaces between the number and the unit of measurement should be introduced throughout the text.

Please introduce all abbreviations the first time they are used in the text (e.g. μCT, BUN, 3pt).

Methods section should be rewritten for clarity. For example, line 94 – 95 “determine various blood gas parameters, including pH, bicarbonate (HCO3-), calcium (Ca2+), potassium (K+), and sodium (Na+)” – in this version this sentence suggests that the pH, bicarbonate, calcium, etc. are blood gas parameters. Moreover, line 114 “(as determined via μCT, section 2.5)” – sections are not numbered.

3. In section Raman spectroscopy of bone, lines 122 – 124 “Five point measurements…surface of the midshaft.” Was this measurement preceded with any bone preparation, such as removal of periosteum? Please include this information.

4. Is there a particular reason for using only 4 left femurs in Raman spectroscopy?

5. What algorithm was used for baseline correction of the raw Raman spectra? Please include this information in the Methods.

6. Is the carbonate:phosphate ratio determined from the intensity ratio of those peaks rather than peak areas?

7. Please include more information on acquisition parameters of the μCT beside pixel size, such as use of filters, scan rotation range, step size, frame averaging used. Moreover, which thresholding method was used in the analysis of the acquired data.

Please make sure that all of the Figure references in the text actually correspond with the stated Figure panels (Lines 250-254, Figure 8).

8. Line 139 - 142. This sentence requires careful revision it implies that the same femurs used in Raman spectroscopy investigations are used in the μCT measurements. However, N given at the end of the sentence is 8. Please clarify the number of samples used for each individual analytical method.

9. This work would benefit significantly by inclusion of analysis of osteoclastic activity. Histological investigations using tartrate-resistant acid phosphatase (TRAP) staining or even evaluation of serum markers of bone resorption (TRAP, CTX-1). Please consider the inclusion of such investigations in future submission.

6. PLOS authors have the option to publish the peer review history of their article (what does this mean?). If published, this will include your full peer review and any attached files.

Reviewer #1: No

Reviewer #2: No

---

## [Author Response · Author response to Decision Letter 0]

30 Aug 2023

Reviewer #1

The article by Moody et al. reports a study on the role of bisphosphonates (BPP) (alendronate) in the recovery of acid-induced bone dissolution. With this aim the authors used a diet-induced murine model of metabolic acidosis (control vs acidosis vs acidosis+BPP; 8 mice / experimental group), to reproduce acid-induced bone loss in patients with chronic kidney disease, and they further characterized structure, mechanics, and composition of bones.

The data reported suggest that BPP, instead of reducing bone loss, induce a more severe acidosis, probably due to reduced osteoclast activity, preventing the release of bicarbonate and other buffering agents from bone.

The study is well conceived, and the reported data support the conclusions, even if they need to be further validated by specific studies focused on the mechanisms regulating the BPP-induced acidosis and on the impact of BPP on osteoclasts activity. The introduction clearly defines the biomedical question; the methods are clear, and figures are generally clear.

I have some comments on Raman experiments and associated results.

- in the manuscript it is not specified how bone samples have been prepared before Raman measurements. Were bone samples cut into sections? In case, which is the thickness, and which is the method used for cutting? Did author perform fixation other than the freezing step?

Thanks for this comment. The bones used for the mechanical testing were then used for external and internal Raman measurements. After mechanical testing, the femoral samples were wrapped in 1X PBS-soaked gauze and frozen at -20°C. Samples were then thawed and measurements were made on the external periosteal surface. The samples were then rotated to show the fracture surface and internal measurements were done on the exposed internal bone. For these measurements, the bones were not sectioned using OCT or paraffin. Rather, the samples were placed in the Raman such that the fracture surface with the exposed internal bone was facing the laser. Further clarification and information were added on lines 127-130 in the “Raman spectroscopy of bone” section. It reads: 

“The same femurs used for exterior measurements were used for interior measurements. Samples were not sectioned, but rather were turned 90° so that the internal surface exposed at the fracture site was facing the Raman laser.”

- I would add the type of grating used for the Raman measurements

The grating used was added to section “Raman spectroscopy of bone” on line 121. It now reads: 

“The composition of the femoral samples was measured using a Witec alpha 300 Raman spectrometer (Witec, Ulm, Germany) with a 785 nm laser at a grating of 300 g/mm.”

- considering that the original measurements included spectral information from 200 and 1800 1/cm, I would not exclude to show the region between 200 and 960 1/cm, including two important phosphate bands, around 430 and 600 1/cm

Thanks for this suggestion. Fig. 5A was changed so the Raman spectra includes information from 200 to 1800 cm-1.

- The use of mineral to matrix and carbonate to phosphate ratio are meaningful. At the same time, I would also consider crystallinity (inverse of the full-width at half maximum (FWHM) of the ν1PO4 (phosphate) peak at around 960 cm-1) as a parameter of bone quality. This could eventually provide interesting insights. From visual examination, a small difference can be appreciated at that level. If needed, here two good references: 1) https://doi.org/10.1007%2Fs11999-010-1692-y 2) https://doi.org/10.1039/D1AN01560E

Thank you for this suggestion and the references. Although no significant changes were seen in the inverse of FWHM using Raman spectroscopy, this data was added to Fig 5 and the manuscript. These references were also added to the manuscript on line 142. 

Reviewer #2

The authors point-out and demonstrate the importance of careful consideration of the acidosis status of CKD patients when prescribing bisphosphonate therapy. While the negative of bisphosphonates on acidosis has been demonstrated, the manuscript would benefit from a major rewriting and additional investigations. Major revision is necessary.

Below is the list of concerns and suggestions for the authors in no particular order of priority.

1. Use of parametric tests in the statistical analysis with a small sample size (N=8 / group) should be avoided. Data should be re-analysed and non-parametric alternative analyses performed. Reporting of P values less than and equal to 0.1 in the graphs is misleading, especially in cases where both the significant and non-significant differences are reported. I suggest those to be removed.

We appreciate your suggestion. However, after talking to the statistical consultant at the University of Connecticut (Dr. Timothy Moore), it was determined that parametric tests can still be done on some of the data. The blood gas data along with the food consumption, urine pH, weight, and fluid consumption were determined to have normal distributions using QQ plots; therefore, this data can utilize parametric statistical analysis. Two-way repeated measures AVONAs and mixed-effect analysis were done on this data. On the other hand, the compositional, mechanical, and structural data did not have enough data to determine normality, so non-parametric tests (in this case, Kruskal-Wallis tests) were used. The Methods sections on lines 177-182 and figures were updated to reflect these changes. It reads: 

"Due to the normal distribution of data, parametric statistical testing using two-way ANOVAs or mixed-effects analysis with post-hoc Bonferroni’s or Tukey’s multiple comparisons tests were used for the blood metrics, food consumption, urine pH, weight, and fluid consumption. For the compositional, mechanical, and structural data, non-parametric statistical testing was used including Kruskal-Wallis tests with post-hoc Dunn’s multiple comparisons tests.”

Details on statistical testing were added to figure descriptions. 

P-values less than and equal to 0.1 are no longer reported in graphs. 

2. Spaces between the number and the unit of measurement should be introduced throughout the text.

Please introduce all abbreviations the first time they are used in the text (e.g. μCT, BUN, 3pt).

Methods section should be rewritten for clarity. For example, line 94 – 95 “determine various blood gas parameters, including pH, bicarbonate (HCO3-), calcium (Ca2+), potassium (K+), and sodium (Na+)” – in this version this sentence suggests that the pH, bicarbonate, calcium, etc. are blood gas parameters. Moreover, line 114 “(as determined via μCT, section 2.5)” – sections are not numbered.

Thank you for your comments. Spaces have been added between the numbers and their corresponding units. Descriptions of abbreviations have been added at their first use. Methods have been changed to add more clarification. The “(as determined via uCT, section 2.5)” has been removed. 

3. In section Raman spectroscopy of bone, lines 122 – 124 “Five point measurements…surface of the midshaft.” Was this measurement preceded with any bone preparation, such as removal of periosteum? Please include this information.

Thanks for this question. More details about bone preparation preceding Raman data acquisition were added to lines 124-125: 

“Before taking measurements, the periosteum was removed using a scalpel blade and fine-grained sandpaper.”

4. Is there a particular reason for using only 4 left femurs in Raman spectroscopy?

Four left femurs were chosen for Raman spectroscopy so the same femurs used for mechanical testing and microcomputed tomography would also be used for Raman. Additionally, a power analysis based on data from a previous in vivo acidosis study [1], a significance of α=0.05, and β=0.08 indicated that a sample size of four is sufficient to measure changes in Raman spectroscopy data (CO3/PO4, Mineral/Matrix, 960 FWHM) on the order of what has been previously measured on bone from mice under our acidosis induction method. 

5. What algorithm was used for baseline correction of the raw Raman spectra? Please include this information in the Methods.

The shape function on the Witec Project program was used for the baseline correction. This is a rolling ball method for background correction, and we used a ball size of 100 wavenumbers to subtract the background from raw Raman spectra. This was information was added to the Methods section on line 134-136:

“The spectra were then cropped to 200-1800 cm-1, background corrected via a rolling ball method with a ball size of 100 wavenumber, and cleared of cosmic rays using the Witec Program 5.3 software.” 

6. Is the carbonate:phosphate ratio determined from the intensity ratio of those peaks rather than peak areas?

Thanks for the question. Even though it is possible to use the intensity ratio for the carbonate:phosphate ratio, it is not the most accurate measurement as it does not account for changes in mineral crystallinity or variations in the molecular environment. By using the peak areas, we account not only for the number of PO43- or CO32- in the ideal crystalline configuration but for the total amount of each moiety in the samples. 

7. Please include more information on acquisition parameters of the μCT beside pixel size, such as use of filters, scan rotation range, step size, frame averaging used. Moreover, which thresholding method was used in the analysis of the acquired data.

Please make sure that all of the Figure references in the text actually correspond with the stated Figure panels (Lines 250-254, Figure 8).

A Scanco 50 microcomputed tomography system was used for the microCT analysis. The scans were run using a CuKα X-ray source with an energy of 55kV and a current of 145 μA. A total angular range of 180o was used with a step size of 0.36o (500 projections) and an acquisition time of 500 ms. No filters or frame averaging were used. This additional information has been added in lines 150-153:

“The scans were run at an energy of 55 kV and current of 145 μA with a Cu Kα X-ray source. Scans were done over an angular range of 180o with a step size of 0.36o (500 projections) and an acquisition time of 400 ms to obtain a 16 μm voxel size. “ 

Thresholding values were selected by an experienced user after preliminary examination of the whole data set. This threshold was globally set for all samples. More details about this were added to lines 164-168:

“Using the Bruker CTan software, both the cortical and trabecular segmentation procedures included global thresholding, despeckling black spots less than 10 pixels from the images, ROI shrink-wrapping, and morphological operations. Thresholding values were selected by an experienced user after a preliminary examination of the data set and set to equal values across all samples. “ 

Figure references for Figure 8 under the Results section have been corrected. 

8. Line 139 - 142. This sentence requires careful revision it implies that the same femurs used in Raman spectroscopy investigations are used in the μCT measurements. However, N given at the end of the sentence is 8. Please clarify the number of samples used for each individual analytical method.

Thanks for this comment. You are correct in that the same femurs that were used from Raman spectroscopy were then used for microcomputed tomography testing. Eight femurs were used from mechanical testing. Out of those eight, four femurs were randomly chosen for Raman spectroscopy. Then the original eight samples were used for microcomputed tomography. The manuscript was changed on lines 122 and 147 to clarify this. 

9. This work would benefit significantly by inclusion of analysis of osteoclastic activity. Histological investigations using tartrate-resistant acid phosphatase (TRAP) staining or even evaluation of serum markers of bone resorption (TRAP, CTX-1). Please consider the inclusion of such investigations in future submission.

Thank you for this suggestion. Although the addition of histological investigation for osteoclastic activity would be beneficial, it is beyond the scope of this study. The focus of this manuscript is to see how bisphosphonates affect metabolic acidosis and bone function. We first wanted to see how the combination of acidosis and bisphosphonates impact the structure, composition, and mechanics of the bone before diving deeper into the mechanisms of bone dissolution under these conditions. In future studies, we plan to use histological (TRAP, toluidine blue) and serum biomarker analysis (CTX, P1NP, PTH) to determine osteoclast and osteoblast number and activity. 

1. Moody, M., et al., Physiochemical Dissolution Governs Early Modifications in Acid-Exposed Murine Bone with Long-term Recovery. Orthopedics and Rheumatology, 2023. 22(1): p. 1-15.

---

## [Decision Letter · Decision Letter 1]

4 Sep 2023

Administration of alendronate exacerbates ammonium chloride-induced acidosis in mice

PONE-D-23-12812R1

Dear Dr. Deymier,

We’re pleased to inform you that your manuscript has been judged scientifically suitable for publication and will be formally accepted for publication once it meets all outstanding technical requirements.

Kind regards,

Furqan A. Shah

Academic Editor

PLOS ONE

Additional Editor Comments (optional):

Reviewers' comments:

Reviewer's Responses to Questions

**Comments to the Author**

1. If the authors have adequately addressed your comments raised in a previous round of review and you feel that this manuscript is now acceptable for publication, you may indicate that here to bypass the “Comments to the Author” section, enter your conflict of interest statement in the “Confidential to Editor” section, and submit your "Accept" recommendation.

Reviewer #1: All comments have been addressed

Reviewer #2: (No Response)

2. Is the manuscript technically sound, and do the data support the conclusions?

Reviewer #1: Yes

Reviewer #2: Yes

3. Has the statistical analysis been performed appropriately and rigorously? 

Reviewer #1: Yes

Reviewer #2: Yes

4. Have the authors made all data underlying the findings in their manuscript fully available?

Reviewer #1: Yes

Reviewer #2: Yes

5. Is the manuscript presented in an intelligible fashion and written in standard English?

Reviewer #1: Yes

Reviewer #2: Yes

6. Review Comments to the Author

Reviewer #1: (No Response)

Reviewer #2: Thank you to the authors for addressing the required revisions and providing further explanations in their replies. I would like to bring attention to a minor inconsistency in the response to question #7. In the response, the authors describe a 500 ms acquisition time for μCT scans, whereas in the manuscript, the acquisition time is reported as 400 ms. Once this discrepancy is addressed, I am confident that the manuscript will meet the standards for publication.

7. PLOS authors have the option to publish the peer review history of their article (what does this mean?). If published, this will include your full peer review and any attached files.

Reviewer #1: No

Reviewer #2: No

---

## [Editor Report · Acceptance letter]

7 Sep 2023

PONE-D-23-12812R1 

Administration of alendronate exacerbates ammonium chloride-induced acidosis in mice 

Dear Dr. Deymier:

I'm pleased to inform you that your manuscript has been deemed suitable for publication in PLOS ONE. Congratulations! Your manuscript is now with our production department. 

Kind regards, 

on behalf of

Dr. Furqan A. Shah 

Academic Editor

PLOS ONE